# Effect of Wakame and Carob Pod Snacks on Non-Alcoholic Fatty Liver Disease

**DOI:** 10.3390/nu11010086

**Published:** 2019-01-04

**Authors:** Daniel Rico, Ana Belén Martin-Diana, Arrate Lasa, Leixuri Aguirre, Iñaki Milton-Laskibar, Daniel Antonio de Luis, Jonatan Miranda

**Affiliations:** 1Department of Research and Technology, Agrarian Technological Institute of Castilla and Leon (ITACyL), Government of Castilla and Leon, Ctra. de Burgos Km. 119, 47071 Valladolid, Spain; ricbarda@itacyl.es (D.R.); mardiaan@itacyl.es (A.B.M.-D.); 2Nutrition and Obesity Group, Department of Nutrition and Food Science, Faculty of Pharmacy, University of Basque Country (UPV/EHU) and Lucio Lascaray Research Center, 01006 Vitoria, Spain; leixuri.aguirre@ehu.eus (L.A.); inaki.milton@ehu.eus (I.M.-L.); jonatan.miranda@ehu.eus (J.M.); 3CIBEROBN Physiopathology of Obesity and Nutrition, Institute of Health Carlos III (ISCIII), 01006 Vitoria-Gasteiz, Spain; 4Endocrinology and Nutrition Department, Hospital Clínico Universitario de Valladolid-IEN, Facultad de Medicina, Universidad de Valladolid, 47005 Valladolid, Spain; dadluis@yahoo.es

**Keywords:** wakame, carob, snack, steatohepatitis, NAFLD

## Abstract

Snacks combining different functional ingredients could represent a useful therapeutic strategy against NAFLD. The present study aimed to analyze the effect of two snack formulations based on carob and wakame flour in the treatment for NAFLD in rats. For this purpose, metabolic syndrome was induced in 50 adult rats by a high-fat high-fructose diet over eight weeks. After this period, rats were fed either normal calorie diets supplemented or not with snack A (1/50 wakame/carob pod) and snack B (1/5 wakame/carob pod) for four additional weeks. After sacrifice, liver composition and serum parameters were analyzed. Different pathways of triacylglycerol metabolism in liver were studied including fatty acid oxidation, fatty acid synthesis, triglyceride assembly and release, fatty acid uptake and glucose uptake. Oxidative stress was also measured. Snack treatment, and mainly B snack, reduced liver triacylglycerol levels by increasing fat oxidation. Moreover, this snack reduced oxidative stress. Therefore, this snack formulation could represent an interesting tool useful for fatty liver treatment.

## 1. Introduction

Non-alcoholic fatty liver disease (NAFLD) has become the most common chronic liver disease worldwide. The incidence of this health alteration has increased within obesity and type 2 diabetes. Furthermore, many authors consider it as the hepatic manifestation of the metabolic syndrome (MetS) [1].

The exact prevalence of the NAFLD is unknown because it is generally asymptomatic, but it is estimated that up to 35% of the western population presents it. From this group of people, approximately 10% will develop non-alcoholic steatohepatitis, a condition with significantly higher risk of mortality due to its potential development on cirrhosis and liver cancer [2,3].

The risk factors for NAFLD include central-type obesity, which is usually associated to insulin resistance. This metabolic situation exerts a negative impact since it causes, due to an increase of lipolysis in the adipose tissue, an increase in the fatty acid (FA) flux from this tissue to liver, contributing to an increase in triglyceride (TG) hepatic production. In addition, high glucose concentrations promote the hepatic TG synthesis, while high insulin concentrations reduce hepatic apolipoprotein B-100 production and the FA oxidation. As a result, TG accumulation in the liver occurs.

Other factors such as genetics or gut microbiota may also promote this disease, but diet should be considered as one of the major contributors. For instance, it has been demonstrated that high fructose consumption, an important component of the western diet, triggers an increase in hepatic de novo lipogenesis, dyslipidemia, insulin resistance, and obesity with central fat deposits [4].

Nowadays, the therapeutic strategy against NAFLD focuses on facing the etiological factor, combining pharmacological treatment with changes in eating habits and lifestyle. The usage of functional foods represents another option, since there are many ingredients that to could help in fatty liver treatment. Studies with algae, brown ones like wakame (*Undaria pinnatifida*) among them, have shown potential to be used on therapies against fatty liver [5,6,7]. On the other hand, several investigations carried out in recent years have postulated carob tree (*Ceratonia siliqua* L.) as an interesting pulse due to its protective effect at hepatic level, mainly through oxidative stress modulation pathways [8,9].

The aim of the present study was to analyze the usefulness of snack formulations based on carob and wakame flour in the treatment of NAFLD in a model of metabolic syndrome induce by feeding.

## 2. Materials and Methods

### 2.1. Ethical Approval

The animal experiment was approved in accordance with the Spanish regulation on animal experimentation (approval no M20_2017_003). All animal experimental protocols were reviewed and approved by the ethics committee on animal welfare of our institution (Comité Ético de Experimentación Animal de la Universidad del País Vasco, CEEA-UPV/EHU).

### 2.2. Animals, Diets, and Experimental Design

The experiment was conducted using 50 six-week-old male Wistar rats (Envigo, Barcelona, Spain). Rats were housed in polycarbonate metabolic cages (Tecniplast Gazzada, Buguggiate, Italy) and placed in an air-conditioned room (22 ± 2 °C) with a 12-h light-dark cycle. After a six-day adaptation period, in order to induce MetS, rats were fed a high-fat high-fructose (HFHF) diet (OpenSource Diets, Denmark; Ref. D09100301), for eight weeks. After this period, randomly 10 rats (randomly selected) were sacrificed after a 12 h of fasting under anesthesia (chloral hydrate) by cardiac exsanguination (MS group). The rest of the animals were randomly distributed into four experimental groups (*n* = 10) and fed a semi-purified normal-caloric diet (3.9 kcal/g diet) for four weeks: the wheat control group (WC), the oat control group (OC), the snack A group (SA) and the snack B group (SB) (Appendix A). Snack A contained 0.1% wakame and 5% carob pod in oat and wheat flour dough. Snack B had the same composition but with 1% of wakame and 5% of carob pod. WC diet represented the standard normal-caloric diet for the treatment of metabolic syndrome in human. It was prepared according to previous studies from our laboratory with rodents [10], as well as to recommendations of the American Institute of Nutrition (AIN-93M) [11]. Considering that oat flour was present in snack formulation, starch for OC diet came from oat and wheat flour in similar amounts. Body weight and food intake were measured daily. At the end of the total experimental period (12 weeks), rats from the five experimental groups were sacrificed after a 12 h of fasting under anesthesia (chloral hydrate) by cardiac exsanguination. Serum was obtained from blood samples after centrifugation (1000 *g* for 10 min, at 4 °C). Livers were dissected, weighed, and immediately frozen in liquid nitrogen. All samples were stored at −80 °C until analysis.

### 2.3. Serum Parameters

Commercially available spectrophotometric kits were used for total cholesterol, HDL-cholesterol, non-esterified fatty acids (NEFA) (Biosystems, Barcelona, Spain), TG (Spinreact, Barcelona, Spain) and transaminases (Spinreact, Barcelona, Spain) determinations. Non-HDL-cholesterol was calculated by subtracting the quantity of HDL-cholesterol to total cholesterol.

### 2.4. Liver Composition

Total liver lipids were extracted according to the method described by Folch et al. [12]. The lipid extract was dissolved in isopropanol and the TG and phospholipid content were measured using a commercial kit (Spinreact, Barcelona, Spain). Protein contents was analyzed using the Lowry method [13] and water content was measured gravimetrically by drying samples at 105 °C until a constant weight was reached.

### 2.5. Oxidative Stress Determinants

Lipid peroxidation was determined spectrophotometrically by measuring the formation of thiobarbituric acid reactive species (TBARS) in liver homogenates using a commercial kit (TBARS Assay Kit, Cayman Chemical Company, Ann Arbor, MI, USA). The TBARS concentrations on the samples were calculated using a standard curve obtained with malondialdehyde (MDA). Superoxide dismutase (SOD) was assessed spectrophotometrically, using a commercial kit (Superoxide Dismutase Activity Assay Kit, BioVision Incorporated, Milpitas, CA, USA).

### 2.6. Enzyme Activities

The activity of the lipogenic enzyme fatty acid synthase (FAS) was measured by spectrophotometry, as previously described [14]. Briefly, liver samples (0.5 g) were homogenized in 5 mL of buffer (150 mM KCl, 1 mM MgCl2, 10 mM N-acetyl cysteine and 0.5 mM dithiothreitol) and centrifugated to 100,000*g* for 40 min at 4 °C. The supernatant fraction was used for FAS activity determination, as the rate of malonyl CoA dependent NADH oxidation [15]. Results were expressed as nanomoles of reduced nicotinamide adenine dinucleotide phosphate (NADPH) consumed per minute per milligram of protein. Liver pyruvate kinase activity was measured by fluorimetry, according to the manufacturer’s protocol (BioVision, Milpitas, CA, USA).

As far as oxidative enzymes are concerned, carnitine palmitoyltransferase-1a (CPT-1a) activity was measured spectrophotometrically in the mitochondrial fraction as previously described [16]. The activity was expressed as nanomoles of coenzyme A formed per minute per milligram of protein. Citrate synthase (CS) activity was assessed spectrophotometrically following the Srere method [17], by measuring the appearance of free CoA. Briefly, frozen liver samples were homogenized in 25 vol (w/v) of 0.1 M Tris-HCl buffer (pH 8.0). Homogenates were incubated for 2 min at 30 °C with 0.1MTris-HCl buffer containing 0.1 mMDTNB, 0.25 Triton X-100, 0.5 mMoxalacetate and 0.3 mM acetyl CoA, and readings were taken at 412 nm. Then, the homogenates were re-incubated for 5 min and readings were taken at the same wavelength. CS activity was expressed as CoA nanomoles formed per minute, per milligram of protein. The protein content of the samples was determined by the Bradford method [18], using bovine serum albumin as standard.

In order to assess the assembly and secretion of very low-density lipoproteins by the liver, microsomal TG transfer protein (MTP) activity was determined fluorometrically by using a commercial kit (Sigma-Aldrich, St. Louis, MI, USA). MTP activity was expressed as percentage of transference.

### 2.7. RNA Extraction and Real Time RT-PCR

RNA samples from 100 mg of hepatic small lobe were extracted using Trizol (Invitrogen, Carlsbad, CA, USA), according to the manufacturer’s instructions. After RNA purity verification, samples were then treated with DNase I kit (Applied Biosystems, California, USA) to remove any contamination with genomic DNA. 1.5 µg of total RNA of each sample was reverse-transcribed to first-strand complementary DNA (cDNA) using iScriptTM cDNA Synthesis Kit (Bio-Rad, Hercules, CA, USA).

Relative mRNA levels of *cpt1a*, *cox2*, *nduf*, *cs*, *ucp2*, *pparα*, *tfam*, *slc2a5*, *slc2a2*, *pnpla*, *lipe*, *cd36*, *acsl1*, *acsl5* and *18 s* were quantified using real-time PCR with an iCyclerTM - MyiQTM real time PCR detection system (BioRad, Hercules, CA, USA). For *cpt1a*, *cox2*, *nduf*, *cs*, *ucp2, pparα*, and *tfam* SYBR Green Master Mix (Applied Biosystems, Foster City, CA, USA) was used. The upstream and downstream primers and probe (TibMolbiol, Berlin, Germany, Eurogentec, Liège, Belgium and Metabion, Munich, Germany) were previously described [19,20]. *fatp2*, *fatp5*, *cd36*, *acsl1*, *acsl5*, *pk*, *fasn*, *dgat1*, *dgat2*, *chreb*, *srebf*, *acaca*, *lxrα*, *pnpla*, *lipe*, and *1 8s* were measured by TaqMan^®^ Gene Expression Assays (Rn00581971_m1, Rn00577177_m1, Rn00580728_m1, Rn00563137_m1, Rn00586013_m1, Rn01455286_m1, Rn00569117_m1, Rn00584870_m1, Rn01506787_m1, Rn00591943_m1, Rn01495769_m1, Rn00573474_m1, Rn00581185_m1, Rn01479965_g1, Rn00689222_m1, Rn03928990_g1) and TaqMan^®^ fast advanced master mix (Applied Biosystems, Foster City, CA, USA). RT-PCR parameters used were those defined by manufacturer’s. 18 s mRNA levels were similarly measured and served as the reference gene. All gene expression results were expressed as fold changes of threshold cycle (Ct) value relative to controls using the 2^−ΔΔCt^ method [21].

### 2.8. Western Blot

Liver samples (100 mg) from the small lobe were homogenized in 1000 μL of cellular PBS buffer (pH 7.4), containing protease inhibitors (100 mM phenylmethylsulfonyl fluoride and 100 mM iodoacetamide). Then, homogenates were centrifuged (800*g* for 10 min at 4 °C) and protein concentrations measured by Bradford method [18], using bovine serum albumin as standard.

Immunoblot analyses were carried out using 60 μg of liver protein extracts which were separated by electrophoresis in precast 7.5% SDS-polyacrylamide gels (Bio-Rad, Hercules, CA, USA) and transferred to PVDF membranes (Merck Millipore, Cork, IRL). The membranes were then blocked with 5 % caseine PBS-Tween buffer for 2 h at room temperature and, subsequently, blotted with the appropriate antibodies overnight at 4 °C. Protein levels were detected via specific antibodies for FATP2 (1:1000), GLUT5 (1:1000) (Santa Cruz Biotech, Dallas, TX, USA), AMPK (1:1000), FATP5 (1:1000) (LifeSpan BioScience, Seattle, WA, USA), GLUT2 (Thermo Fisher Scientific, Rockford, IL, USA), and β-Actin (1:5000) (Sigma, St Louis, MO, USA). Afterward, polyclonal anti-mouse for GLUT5 and β-Actin (1:5000) (Santa Cruz Biotech, Dallas, TX, USA), anti-goat for FATP2 (1:5000) (Santa Cruz Biotech, Dallas, TX, USA) and anti-rabbit for FATP5 and GLUT2 (1:5000) (Santa Cruz Biotech, Dallas, TX, USA) were incubated for 2 h at room temperature, and the levels of the afore mentioned proteins were measured. The bound antibodies were visualized by an ECL system (Thermo Fisher Scientific Inc., Rockford, IL, USA) and quantified by a ChemiDoc MP Imaging System (Bio-Rad, Hercules, CA, USA). Specific bands were identified by using a standard loading buffer (Precision Plus protein standards dual color; ref. 161-0374 Bio-Rad).

### 2.9. Statistical Analysis

Results are presented as mean ± SEM. Statistical analysis was performed using SPSS 21.0 (SPSS, Chicago, IL, USA). All the variables were normally-distributed according to the Shapiro–Wilks test. Data were analyzed by one-way ANOVA followed by Newman–Keuls post hoc test. Significance was assessed at the *p* < 0.05 level.

## 3. Results

### 3.1. Effect of the Snacks on Liver Weight and Liver Composition

Table 1 shows body weight, liver weight and liver composition of control and snack-treated rats. When comparing rats showing MetS with those fed the different experimental diets, even though no changes in body weight were found, liver weight/body weight ratio was significantly lower in rats fed the wheat and oat control diets, as well as in rats treated with both snacks when compared to the control group. Liver composition analysis revealed that both control and snack-treated groups had lower TG and phospholipid content in liver than not treated rats showing MetS. Moreover, in the case of SB rats, the percentage of liver TG was even lower than that of WC group. Liver protein percentage remained unchanged with any of the control diets although an increase was observed after snack A consumption. Liver water content increased in OC rats treated and remained unchanged after snack treatment.

### 3.2. Effect of the Snacks on Serum Parameters

When serum parameters were analyzed, MS rats showed lower TG values than the rats fed the control diets (Table 2). However, snack-treatment (A and B) reduced serum TG concentration when compared to control diet fed rats and reached the same level than that of MS rats. While control diets did not modify serum NEFA levels, both snacks effectively reduced them. Regarding to plasma cholesterol levels, although total and HDL-cholesterol values were similar among all the experimental groups, non-HDL cholesterol levels were lower in all groups compared to MS. However, no snack-induced effect was observed on this parameter. As far as plasma transaminases (GOT and GPT) are concerned, neither snack A nor snack B modified these parameters comparing to MS or control groups.

### 3.3. Effect of the Snacks on Oxidative Stress

In relation to oxidative stress related parameters, treatment with both control diets reduced TBARS levels. In the case of OC diet, dropped SOD activity was also appreciated (Figure 1A,B). in the same line, both snacks reduced TBARS levels when comparing to MS group, although no changes were observed when comparing to control groups. SOD activity remained unchanged in the case of snack A, while an increase was found after snack B ingestion.

### 3.4. Effect of the Snacks on β-Oxidation Related Gene Expression and Enzyme Activity

Gene expression of genes and transcription factors involved in liver β-oxidation in liver was analyzed (Figure 2A). Oat control diet did not modify any of the analyzed genes, with the exception of *cox2*, which was significantly higher. However, oxidative enzyme activities remained unchanged (Figure 2B,C). Snack A increased *nduf* mRNA levels comparing to MS group (Figure 2A). Snack B treatment increased CPT and CS activities and snack A only that of CS (Figure 2B,C).

### 3.5. Effect of the Snacks on Lipogenesis Related Gene Expression and Enzyme Activity

OC diet did not modify the expression of any of the genes involved in lipogenesis (Figure 3A) or FAS activity (Figure 3B), but reduced PK activity (Figure 3C). Treatment with snack A enhanced lipogenic gene expression when compared to MS group, but without modifying any parameter when compared to OC group. Snack B only increased chreb mRNA levels (Figure 3A). None of the tested snacks modified FAS or PK activities (Figure 3B,C).

### 3.6. Effects of the Snacks on the Expression Of Genes Involved in Lipolysis, Triacylglycerol Re-Esterification and Release

No changes were observed in lipases gene expression among the different experimental groups (data not shown). *dgat1* and *dgat2* gene expressions were measured in order to see the effects on TG assembly. OC diet increased *dgat2* mRNA levels when compared to MS group (Figure 4A). By contrast, no snack-treatment effect was observed. Moreover, MTP activity was measured in order to clarify whether snacks administration reduced TG synthesis in liver through this enzyme, but significant changes were not observed among all the experimental groups (Figure 4B).

### 3.7. Effect of the Snacks on Fatty Acid Uptake Related Gene and Protein Expression

With regard to genes involved in liver fatty acid uptake, only in the case of SB group reduced gene expression of *acsl1* when compared to OC and SA group was observed (Figure 5A). Neither OC diet nor snacks A and B modified protein expression of FATP2 and FATP5 (Figure 5B,C).

### 3.8. Effect of the Snacks on Glucose Uptake Related Gene and Protein Expression

No changes were observed in glut2 and glut5 gene expression or protein level (Figure 6A–C)

## 4. Discussion

The present study aimed to analyze the potential beneficial effects of new snacks, formulated by using wakame and carob pod, in the treatment of liver steatosis in rats showing MetS. For this purpose, once MetS was induced in rats by a high-fat high-fructose feeding, the animals were then fed a diet in which complex carbohydrates were replaced with the functional snacks. Wakame and carob pod were chosen as functional ingredients for the snack preparation, taking into account the results from our previous research in cell cultures and sensory-panel [22,23]. The WC group included rats fed a standard diet, representing the commonly used dietary strategy of non-healthy dietary patterns normalization. In addition, taking into account the carbohydrate source of the snacks, the OC group was included as a better control group for snack-fed groups.

It has been reported that insulin resistance is one of the main factors involved on the etiopathogenesis of liver steatosis. In fact, NAFLD is considered the hepatic manifestation of MetS [24]. In the present study rats with MetS effectively showed liver mild steatosis because hepatic triglyceride amount was slightly greater than 5% [25]. As expected, the switch from a high-fat high-fructose diet to a standard one (WC or OC) led to a significant reduction in this parameter. The reductions induced by both snacks were higher than those observed in control groups, although without statistical significance. In spite of these changes, transaminase levels remained unchanged. Concerning this issue, it is important to emphasize that although transaminases are commonly used as biomarkers of fatty liver, they are not reliable markers since their levels can be normal even in advanced NAFLD [24,26]. In addition, reductions in liver steatosis without changes in transaminase levels have been described in the literature [27,28].

In order to gain more insight concerning the effects of the functional snacks, the expression of genes involved in hepatic triglyceride metabolism were analyzed. In these measurements, only OC was included as the control group in order to differentiate whether the observed effect was due to the functional ingredients of the snack or to the oat. No significant changes were observed in liver from rats treated with these snacks in the expression of key enzymes that play a key role in de novo lipogenesis, (*acaca* and *fasn*), and *pk*, nor in the expression of transcriptional factors regulating this metabolic process (*srebf1* or *cherb*) when compared to the control groups. In addition, the activities of PK and FAS remained unchanged. A similar situation was observed when gene and protein expressions of fatty acid transporters, such as FATP2 or FATP5, were assessed. Snack treatment did not modify glucose uptake, either. However, when fatty acid oxidation pathway was analyzed, although no changes in gene expression were found, both snacks increased the activity of CTP-1a, when compared to the control groups. Additionally, the snack B, which contained more wakame seaweed percentage, also significantly increased the activity of CS. These results suggest that functional snacks could increase fatty acid oxidation, thus reducing fatty acid availability for TG synthesis. TG assembly, another important process involved in TG synthesis, was also analyzed. Snack B, but not snack A, significantly reduced the gene expression of *dgat1*, when compared to the control groups. Whereas *dgat2* function is closely linked to endogenous fatty acid synthesis and esterification, *dgat1* may be involved in the recycling of hydrolyzed TG by re-esterifying the fatty acids [29]. Thus, this pathway could represent another mechanism by which snack B reduced hepatic fat content.

Our results cannot be compared with those obtained in other studies since the snacks used in the present study have not been previously used. Nevertheless, we can consider reported studies carried out with wakame or carob pod used independently. In this sense, our results are in agreement with those reported by Murata et al. [6]. They found that dietary wakame decreased rat liver TG level due to increased activities of enzymes involved in the fatty acid oxidation in mitochondria. In a more recent study, carried out in growing rats (four weeks old), Yoshinaga et al. showed that 1% of wakame administration was useful for MetS prevention [30]. Through a DNA array, this research proposed that the administration of wakame downregulated the up-stream transcription factor SREBF-1, leading to fatty acid synthesis inhibition, and upregulated the transcription factor PPARα, thus increasing fatty acids transport into mitochondria for β-oxidation and bile acid biosynthesis. Although the purpose of our study and the work conducted by Yoshinaga et al. was different [30], MetS prevention in the case of Toshinaga et al., and MetS treatment in the present study, as well as the maturation state of rats, growing or young-adult rats, fatty acid oxidation was proposed as mechanism of action of wakame in both studies.

With regard to carob pod, a study conducted in rabbits revealed that an eight-week treatment with insoluble fiber obtained from carob pod, rich in polyphenols, prevented dyslipidemia. Supplementation with this polyphenol-rich fiber increased liver expression of SIRT1 and PGC-1α, and modulated key factors involved in hepatic lipid metabolism [31].

The differences observed in TG metabolism among rats fed in functional snacks and control rats do not fit well with the lack of differences observed among these groups in terms of hepatic TG content. A possible explanation could be that the changes induced by the snacks were not big enough to induce a significantly higher reduction on this parameter. Furthermore, it could hypothesized that perhaps a longer period of treatment could enhance the positive effects of the snacks.

Taking into account the results concerning hepatic TG content, it seems that the functional snacks do not represent an advantage with regard to the improvement induced by dietary pattern normalization. Nevertheless, when assessing the treatment of NAFLD, other additional aspects such as oxidative stress should be considered. It has been demonstrated that high-fructose feeding causes fat deposition in the liver and induces oxidative stress, essentially through the deterioration of the antioxidant defense system [32]. Moreover, elevated oxidative stress and subsequent lipid peroxidation have been reported to play a crucial role in the pathogenesis of MetS, and it is well known that they are essential in the progression from NAFLD to steatohepatitis [33,34,35,36].

When oxidative stress was analyzed, it was observed that in rats treated with the snack B SOD values were significantly increased, meaning that it exerted an anti-oxidant effect. Bearing in mind the difference among rats treated with the snack B and the control groups, it can be proposed that wakame seaweed, and more precisely its polyphenols as well as fucoxanthine derived liver metabolites, can be responsible for this anti-oxidant effect [5,37].

One limitation of the present study is that dietary carbohydrates were totally replaced by functional snacks, an experimental condition that cannot be reproduced in human trials. In view of this situation, we are working in an ongoing intervention human study, where subjects with MetS receive 50 g of formulated functional snack per day. Moreover, considering that several studies have shown that female gender is a risk factor for NAFLD [38], the results obtained in this study in male rats cannot be directly extrapolated to females. Finally, it must be pointed out that performing liver immunohistochemistry assays could have given an additional approximation to the NAFLD diagnosis carried out by measuring hepatic TG content.

Altogether, the results obtained show that, under the present experimental conditions, snack B can be considered as an interesting tool for fatty liver treatment because it induces not only a reduction in triglyceride amount, but also a reduction in oxidative stress. Thus, a greater amelioration of NAFLD than that obtained by switching rats to a standard feeding pattern can be achieved by the proposed snack.

## Figures and Tables

**Figure 1 nutrients-11-00086-f001:**
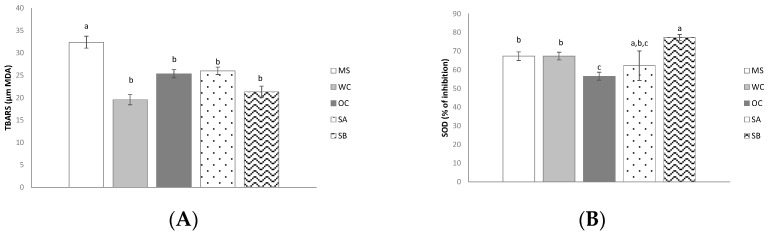
Effects on oxidative stress of MetS rats (MS), rats with fed with wheat control diet (WC, *n* = 10), wheat and oat control diet (OC, *n* = 10), A snack (SA, *n* = 10), and B snack (SB, *n* = 10). (**A**) TBARS activity and (**B**) SOD activity. Values are expressed as mean ± SEM. Values not sharing a common letter are significantly different (*p* < 0.05).

**Figure 2 nutrients-11-00086-f002:**
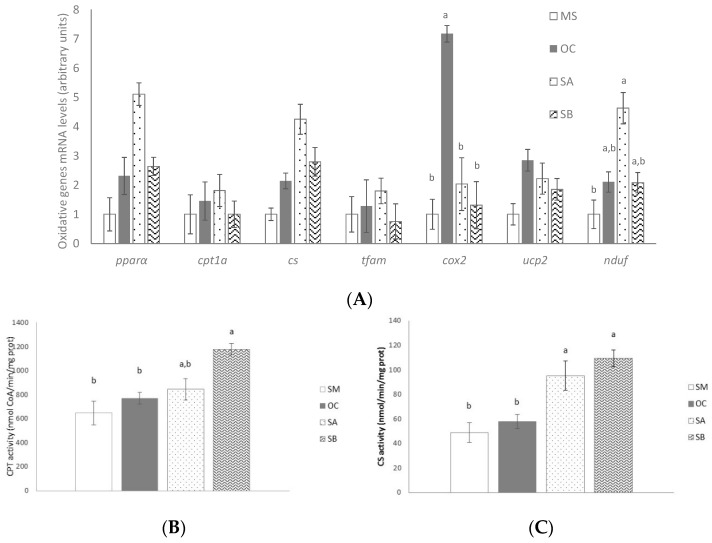
Effects on fatty acid oxidation of MetS rats (MS, *n* = 10), rats fed with wheat control diet (WC, *n* = 10), wheat and oat control diet (OC, *n* = 10), A snack (SA, *n* = 10), and B snack (SB, *n* = 10). (**A**) Oxidative genes mRNA levels, (**B**) CPT activity, and (**C**) CS activity. Values are expressed as mean ± SEM. Values not sharing a common letter are significantly different (*p* < 0.05). In the case of CPT1a and CS activities, *p* value for the comparisons SM and SB groups or OC and SB groups is *p* < 0.001.

**Figure 3 nutrients-11-00086-f003:**
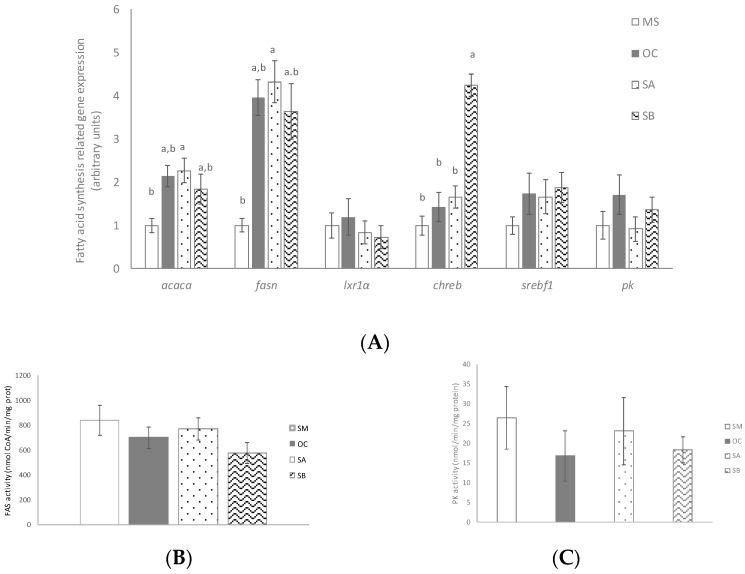
Effects on fatty acid synthesis of MetS rats (MS, *n* = 10), rats fed with wheat control diet (WC, *n* = 10), wheat and oat control diet (OC, *n* = 10), A snack (SA, *n* = 10), and B snack (SB, *n* = 10). (**A**) Fatty acid synthesis involved genes mRNA levels, (**B**) FAS activity and (**C**) PK activity. Values are expressed as mean ± SEM. Values not sharing a common letter are significantly different (*p* < 0.05). In the case of *cherb* gene expression, *p* value for the comparison between MS, OC and SA is *p* < 0.01.

**Figure 4 nutrients-11-00086-f004:**
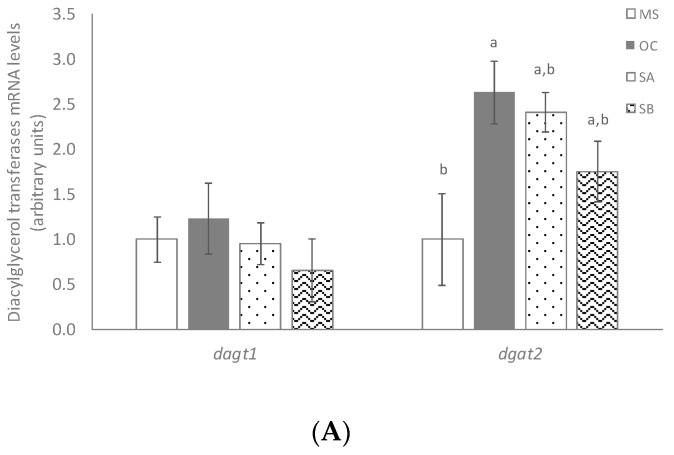
Effects on triacylglycerol re-esterification and triacylglycerol release of MetS rats (MS, *n* = 10), rats fed with wheat control diet (WC, *n* = 10), wheat and oat control diet (OC, *n* = 10), A snack (SA, *n* = 10), and B snack (SB, *n* = 10). (**A**) triacylglycerol re-esterification involved genes mRNA levels and (**B**) MTP activity. Values are expressed as mean ± SEM. Values not sharing a common letter are significantly different (*p* < 0.05).

**Figure 5 nutrients-11-00086-f005:**
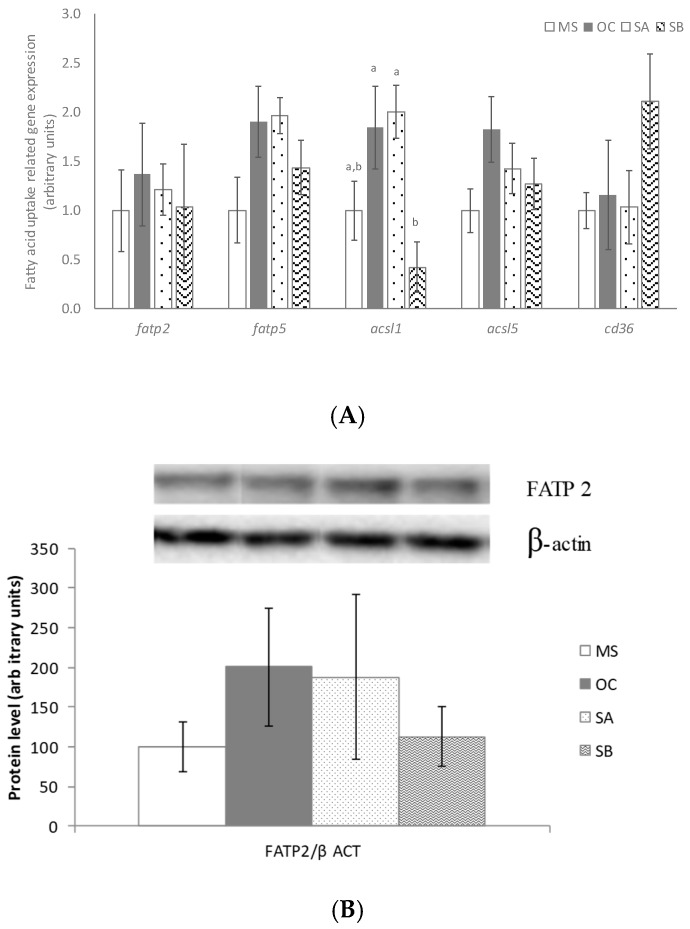
Effects on fatty acid uptake of MetS rats (MS, *n* = 10), rats fed with wheat control diet (WC, *n* = 10), wheat and oat control diet (OC, *n* = 10), A snack (SA, *n* = 10), and B snack (SB, *n* = 10). (**A**) Fatty acid uptake involved genes mRNA levels, (**B**) FATP2 protein levels and (**C**) FATP5 protein levels. Values are expressed as mean ± SEM. Values not sharing a common letter are significantly different (*p* < 0.05). In the case of acsl1 gene expression, *p* value for the comparison between OC and SA vs. SB is *p* < 0.01.

**Figure 6 nutrients-11-00086-f006:**
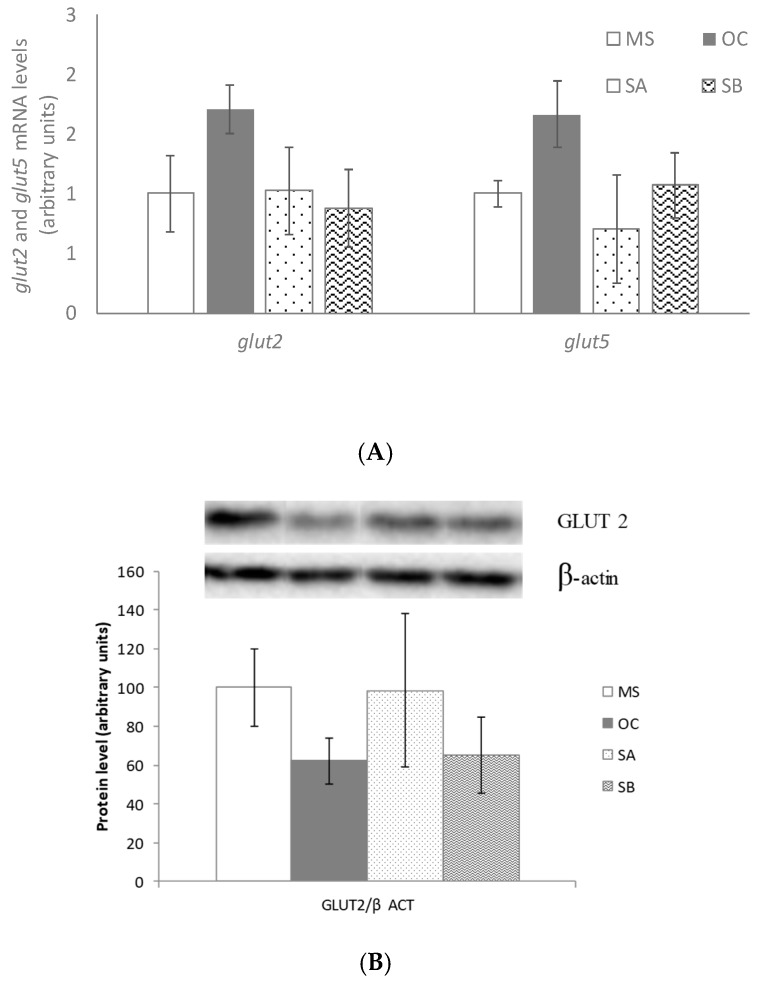
Effects on glucose uptake of MetS rats (MS, *n* = 10), rats fed with wheat control diet (WC, *n* = 10), wheat and oat control diet (OC, *n* = 10), A snack (SA, *n* = 10), and B snack (SB, *n* = 10). (**A**) Glucose uptake involved genes mRNA levels, (**B**) GLUT2 protein levels and (**C**) GLUT5 protein levels. Values are expressed as mean ± SEM. Values not sharing a common letter are significantly different (*p* < 0.05).

**Table 1 nutrients-11-00086-t001:** Final body weight, liver weight, and liver composition of experimental groups.

	MS	WC	OC	SA	SB
Final body weight (g)	412.0 ± 10.0	456.0 ± 14.0	449.0 ± 5.0	441.0 ± 8.0	448.0 ± 12.0
Liver weight (g)	15.5 ± 0.6 ^a^	15.0 ± 0.3 ^a^	14.8 ± 0.7 ^a^	13.9 ± 0.7 ^a^	13.8 ± 0.7 ^a^
Liver/body weight	3.7 ± 0.1 ^a^	3.3 ± 0.1 ^b^	3.3 ± 0.1 ^b^	3.1 ± 0.1 ^b^	3.1 ± 0.1 ^b^
Liver composition:					
Triglycerides (%)	5.31 ± 0.51 ^a^	4.29 ± 0.39 ^b^	3.66 ± 0.41 ^b,c^	3.43 ± 0.29 ^b,c^	3.15 ± 0.35 ^c^
Phospholipids (%)	2.14 ± 0.11 ^a^	2.29 ± 0.15 ^b^	2.10 ± 0.13 ^b^	2.90 ± 0.27 ^b,c^	2.26 ± 0.08 ^c^
Protein (%)	13.2 ± 0.5 ^b^	16.2 ± 1.3 ^a,b^	15.0 ± 0.9 ^a,b^	16.8 ± 0.6 ^a^	16.1 ± 1.1 ^a,b^
Water (%)	61.4 ± 1.0 ^b^	64.0 ± 0.9 ^b^	72.9 ± 0.9 ^a^	66.6 ± 1.2 ^b^	65.3 ± 1.5 ^b^

Values are means ± SEM. Values in the same row with different subscript are significantly different at *p* < 0.05. In the case of water values, different letters mean *p* < 0.001. MS: metabolic syndrome; WC: wheat control; OC: oat control; SA: snack A; SB: snack B.

**Table 2 nutrients-11-00086-t002:** Serum parameters in rats fed on the experimental diets.

	MS	WC	OC	SA	SB
Triacylglycerol (mg/L)	37.1 ± 6.4 ^b^	64.0 ± 22.8 ^a^	46.6 ± 15.4 ^a,b^	37.4 ± 11.0 ^b^	40.6 ± 18.4 ^b^
NEFA (mmol/L)	0.51 ± 0.10 ^a^	0.44 ± 0.12 ^a,b^	0.39 ± 0.09 ^a,b^	0.35 ± 0.04 ^b^	0.29 ± 0.03 ^b^
Total cholesterol (mg)	150.0 ± 23.4	147.1 ± 15.9	143.0 ± 12.0	142.1 ± 19.6	136.7 ± 5.8
HDL-cholesterol (mg)	13.9 ± 4.3 ^a^	10.2 ± 1.9 ^b^	10.7 ± 2.1 ^a,b^	11.2 ± 2.6 ^a,b^	11.1 ± 0.7 ^a,b^
Non-HDL-cholesterol	136.1 ± 23.3 ^a^	136.9 ± 15.7 ^b^	131.2 ± 11.2 ^b^	130.9 ± 20.3 ^b^	125.0 ± 6.0 ^b^
Transaminases					
GPT (U/L)	42.0 ± 4.3	25.9 ± 1.7	29.0 ± 2.1	38.7 ± 6.6	32.4 ± 4.7
GOT (U/L)	124.8 ± 5.3 ^a,b^	109.5 ± 3.3 ^a,b^	127.3 ± 5.9 ^a,b^	155.3 ± 25.3 ^a^	96.2 ± 11.1 ^b^

Values are means ± SEM. Values in the same row with different subscript are significantly different at *p* < 0.05. In the case of NEFA, *p* value for the comparison between MS and SB is *p* < 0.01. NEFA: non sterified fatty acid, HDL: high density liporpotein, ALT: alanine aminotransferase, AST: aspartate aminotransferase. MS: metabolic syndrome; WC: wheat control; OC; oat control; SA: snack A; SB: snack B.

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
