# Peer review of "Effect of Wakame and Carob Pod Snacks on Non-Alcoholic Fatty Liver Disease"

_nutrients, 2019, doi:10.3390/nu11010086_

Reviewer 1 Report

The research article entitled “Effect of wakame and carod pod snacks on Non-Alcoholic Fatty Liver Disease” is systematically written and all data is clearly discussed. Interestingly, authors also included limitations of the study and have come up with formulated functional snacks for human study. It would be a great addition to the manuscript if the author could include few correlative data sets from an ongoing human study if the study has reached any time point. The study is highly significant and could lead a nutraceutical tool to treat NAFLD.

The study is able to justify several metabolic observations however, transaminase levels did not change. As we all aware that transaminases are commonly used biomarker of liver disease, but it is not very specific for NAFLD. Therefore, it is highly suggested to provide liver histopathology data of these study groups. The liver biopsy/histopathology is a still gold standard of NAFLD diagnostics.  

The author analyses hepatic lipid peroxidation in the study groups using liver homogenate. It is important to know that which of the hepatic zone (between hepatic portal vein to central vein) have increased oxidative stress and lipid peroxidation in NAFLD and did formulated snacks were able to decrease the lipid peroxidation in that area. The author should provide Immunohistochemistry of 4-hydroxynonenal (stable product of lipid peroxidation) and compare the data between the groups. The author may refer several publications on hepatic 4-HNE in NAFLD (https://www.ncbi.nlm.nih.gov/pubmed/27913210, https://www.ncbi.nlm.nih.gov/pubmed/24211274, https://www.ncbi.nlm.nih.gov/pubmed/27876107).

Author Response

Reviewer 1

The research article entitled “Effect of wakame and carod pod snacks on Non-Alcoholic Fatty Liver Disease” is systematically written and all data is clearly discussed. Interestingly, authors also included limitations of the study and have come up with formulated functional snacks for human study. It would be a great addition to the manuscript if the author could include few correlative data sets from an ongoing human study if the study has reached any time point. The study is highly significant and could lead a nutraceutical tool to treat NAFLD.

We agree with the referee that including data obtained in the human study concerning the effect of the snack on NAFLD would be great. However, human study has not finished yet and thus, we only have some preliminary data that cannot be published so far. In order to answer to the interest of the referee we can comment in advance that human preliminary data show no effect of the snack in transaminases levels or in several liver indexes such as FLI (Fatty Liver Index) or LAP (Liver Accumulation Product).

Table 1. Preliminary data of liver indexes obtained in the human study.

Parameter

Intervention snack

Control snack

Basal

8 weeks

Basal

8 weeks

FLI

90.9 ± 13.9

92.4 ± 9.5

96.1 ± 3.1

96.2 ± 2.8

LAP

4.5 ± 0.6

    4.5 ± 0.7

    4.7 ± 0.4

    4.7 ± 0.5

As the human study is still ongoing and we have not definitive data yet, no correlation can be performed between data obtained in animals and humans.

The study is able to justify several metabolic observations however, transaminase levels did not change. As we all aware that transaminases are commonly used biomarker of liver disease, but it is not very specific for NAFLD. Therefore, it is highly suggested to provide liver histopathology data of these study groups. The liver biopsy/histopathology is a still gold standard of NAFLD diagnostics.  

The referee is right that the transaminases are not very specific for NAFLD and that liver histopathology data would give a better insight on NAFLD status in rats. Unfortunately, liver samples were not adequately stored for immunohistochemistry assays and thus, we cannot provide the data asked for by the referee.

Nevertheless, in addition to transaminase values, we provide liver TG data, which if higher than 5%, represent liver steatosis (Nassir et al., 2015). It is true that in the present study, liver TG content was in the edge of the 5% needed, and thus, we have changed the sentence in the discussion and have suggested that a “mild” steatosis was achieved in MS group (lines 307 and 308).

In accordance with the referee, we also consider that liver histopathology is still the gold standard of NAFLD diagnosis. For this reason, the fact that no immunohistochemistry assay was performed has been included as another limitation of the present study (lines 376 and 378). 

Nassir, F.; Rector, R.S.; Hammoud, G.M.; Ibdah, J.A. Pathogenesis and Prevention of Hepatic Steatosis. Gastroenterol Hepatol (N Y) 2015, 11, 167-175. 

The author analyses hepatic lipid peroxidation in the study groups using liver homogenate. It is important to know that which of the hepatic zone (between hepatic portal vein to central vein) have increased oxidative stress and lipid peroxidation in NAFLD and did formulated snacks were able to decrease the lipid peroxidation in that area. The author should provide Immunohistochemistry of 4-hydroxynonenal (stable product of lipid peroxidation) and compare the data between the groups. The author may refer several publications on hepatic 4-HNE in NAFLD (https://www.ncbi.nlm.nih.gov/pubmed/27913210, https://www.ncbi.nlm.nih.gov/pubmed/24211274, https://www.ncbi.nlm.nih.gov/pubmed/27876107).

Liver composition and enzymes activities were performed in the small lobe of the liver. This information has been included in the new version of the manuscript (lines 129 and 149).  As stated before, no samples were collected for immunohistochemistry assay and thus authors cannot provide requested hepatic 4-HNE results. We feel that this fact can be considered a weakness of the study. Therefore, this information has been included as a limitation of the research. However, taking into account that 4-HNE is a reliable biomarker of lipid oxidation in liver diseases as NAFLD, the references proposed by the referee have been included in the revised manuscript as evidence of lipid oxidation during liver injury (references 34, 35 and 36).

34. Seth, R.K.; Das, S.; Dattaroy, D.; Chandrashekaran, V.; Alhasson, F.; Michelotti, G.; Nagarkatti, M.; Nagarkatti, P.; Diehl, A.M.; Bell, P.D.; Liedtke, W.; Chatterjee, S. TRPV4 activation of endothelial nitric oxide synthase resists nonalcoholic fatty liver disease by blocking CYP2E1-mediated redox toxicity. Free Radic Biol Med 2017, 102, 260-273. doi: 10.1016/j.freeradbiomed.2016.11.047.

35. Seth, R.K.; Das, S.; Kumar, A.;, Chanda, A.; Kadiiska, M.B.; Michelotti, G.; Manautou, J.; Diehl, A.M.; Chatterjee, S. CYP2E1-dependent and leptin-mediated hepatic CD57 expression on CD8+ T cells aid progression of environment-linked nonalcoholic steatohepatitis. Toxicol Appl Pharmacol 2014, 274, 42-54. doi: 10.1016/j.taap.2013.10.029.

36. Jin, C.J.; Engstler, A.J.; Sellmann, C.; Ziegenhardt, D.; Landmann, M.; Kanuri, G.; Lounis, H.; Schröder, M.; Vetter, W.; Bergheim, I. Sodium butyrate protects mice from the development of the early signs of non-alcoholic fatty liver disease: role of melatonin and lipid peroxidation. Br J Nutr 2016, 23, 1-12.

Reviewer 2 Report

I read this pleasure this paper about the possible use of a wakame/carob pod combination as a possible nutraceutical agent in NAFLD. 

The Introduction is satisfactory, the Methods have been carefully explained and the Results have been clearly reported. In the Discussion, there are a number of factors that have not been taken into account (especially limitation in the possbile translational aspects from rodents to humans.

These aspects should be better discussed to improve the quality of the manuscript. In detail:

1) METHODS - ANIMALS, DIETS AND EXPERIMENTAL DESIGN. I think that a figure (flow chart) illustrating the various step of the experimental design would help the readers.

2) The choice of the mice models and the diet is correct, in my opinion. As it often happens in these studies, all of the mice are male. While there are reasons favouring this choice, the role of sex in human NAFLD is very important. In this way, the role of sex is neglected, limiting the potential translational applications of the study. Please report the reasons for which you preferred to study only male mice and discuss this limitation in the Discussion

3) From the previous point, in most animal studies the diet is relatively short and the animals are sacrificed after a few weeks. In this way it is possible to study the total fat content of the liver, but it is very difficult to quantify inflammation and fibrosis. These phenomena require a long tome to develop but they are the most important prognosticators of liver-related events in human. Tryglicerides contents alone are not so important. As such, in this moment it is difficult to hyptothesise which subject would most benefit from the snack. Please add some additional considerations in this regard in the Discussion

Author Response

Reviewer 2

1) METHODS - ANIMALS, DIETS AND EXPERIMENTAL DESIGN. I think that a figure (flow chart) illustrating the various step of the experimental design would help the readers.

According to the referee´s comment, a flow chart illustrating the experimental design has been included as supplementary figure 1 in the revised version.

2) The choice of the mice models and the diet is correct, in my opinion. As it often happens in these studies, all of the mice are male. While there are reasons favouring this choice, the role of sex in human NAFLD is very important. In this way, the role of sex is neglected, limiting the potential translational applications of the study. Please report the reasons for which you preferred to study only male mice and discuss this limitation in the Discussion.

We agree with the referee that selecting only male rats for the present study represents a limitation in the translation of the obtained results from rodents to humans. We added this fact as one of the limitations of the study (lines 373 to 375 of the revised version).

The reasons for selecting male rats in the present study were that male rats do not present hormonal variations as female ones do during their ovaric cycle. Moreover, main studies published in the literature are performed in male rats and thus, comparison of the obtained data with that present in the literature is easier when performing experiments in male rats.

3) From the previous point, in most animal studies the diet is relatively short and the animals are sacrificed after a few weeks. In this way it is possible to study the total fat content of the liver, but it is very difficult to quantify inflammation and fibrosis. These phenomena require a long time to develop but they are the most important prognosticators of liver-related events in human. Tryglicerides contents alone are not so important. As such, in this moment it is difficult to hyptothesise which subject would most benefit from the snack. Please add some additional considerations in this regard in the Discussion

We agree with the referee that the analysis of inflammation and fibrosis should give interesting data as they are the most important prognosticators of liver-related disorders in human. In fact, inflammatory markers were analyzed in plasma in the present study but they were included in another manuscript that is under revision. Results obtained were as follows:

A

B

C

Figure 1. Anti-inflammatory effect of functional-diet feeding. Serum levels of C-reactive protein (CRP), monocyte chemoattractant protein 1 (MCP1) and interleukin 6 (IL6) at the end of the experiment in rats fed a wheat control diet (WC, n=10), oat control diet (OC, n=10), snack A diet (SAH, n=10) and snack B diet (SBH, n=10) diet. Values are expressed as mean ± SEM. Data not sharing a common letter are significantly different (P < 0.05).

As explained by the reviewer, inflammation, and mainly fibrosis, are phenomena that require a long time to develop. When the effect of functional ingredients on liver is analyzed, it must be pointed out that these ingredients can show different effects on simple steatosis than on steatohepatitis, being this last disease the one that presents inflammation and fibrosis (Heebøll et al., 2016).

The present study was focused to analyze the effect of the functional snack on simple steatosis and for that purpose the described experimental design was carried out, and no inflammation or fibrosis was analyzed in the liver.

Heebøll S, El-Houri RB, Hellberg YE, Haldrup D, Pedersen SB, Jessen N, Christensen LP, Grønbaek H. Effect of resveratrol on experimental non-alcoholic fatty liver disease depends on severity of pathology and timing of treatment. J Gastroenterol Hepatol. 2016 Mar;31(3):668-75. doi: 10.1111/jgh.13151.

Reviewer 3 Report

Comments to the Author 

The manuscript submitted by Rico et al., evaluated the pharmacological actions of wakame and carod pod snack against the NAFLD.

The authors of the study induced the metabolic syndrome in rats by feeding a high fat high fructose diet. Then, they replaced the diet with the functional snacks. In this experimental set up, the authors were able to identify that snack B used in the study was effective in reducing the TG levels through fatty acid oxidation. They have also reported that the mice fed a snack have reduced oxidative stress than the rats that fed a high fat high fructose diet. Altogether, the results of the current study show that, under the present experimental conditions, the snack B can be considered as an interesting nutraceutical tool for fatty liver treatment.

It is a well written and well-structured manuscript. However there are few instances that require clarification and further experiments.

Major comments: 

1)      The authors mentioned in the materials section (line 66 to 72) that they sacrificed 10 rats at the end of 8weeks of high fat high fructose diet feeding. I assume they have done it to confirm the NAFLD in the rats. If so, the authors need to provide a data (oil redo o and Masson trichome staining) to show that at this time point rats had a NAFLD. This will help in understanding the severity of the disease and potency of the snacks used.

2)      Table 1 shows a body weight increase by the snacks fed rats than the MetS group. Although they mentioned it statistically not significant, an increase of approximately 25-30g in a 4 week period would be a serious concern. Hence it needs to be justified.

3)      Statistical comparison between the groups is confusing. I do not understand what “a” and “b” denotes in all the figures. The authors need to indicate in figure legends

4)      Results presented in Fig 5B-C need to be discussed  

5)      There is no explanation or discussion of the results presented in Fig 6A-C

Author Response

Reviewer 3

Major comments: 

1)      The authors mentioned in the materials section (line 66 to 72) that they sacrificed 10 rats at the end of 8weeks of high fat high fructose diet feeding. I assume they have done it to confirm the NAFLD in the rats. If so, the authors need to provide a data (oil redo o and Masson trichome staining) to show that at this time point rats had a NAFLD. This will help in understanding the severity of the disease and potency of the snacks used.

Unfortunately, Oil Red O and Masson trichome staining were not used in the present study. TG content was performed by Folch method, a widely used method for liver fat quantification (Folch et al., 1957; Lasa et al., 2011; Qasem et al., 2015) and MS group showed, after 8 weeks of HFHF feeding 5.31% of TG in liver. As mentioned in the Discussion section when hepatic triglyceride amount is greater than 5%, it means that there is liver steatosis (Nassir et al., 2015). It is true that in the present study, liver TG content was in the edge of the 5% needed, and thus, we have changed the sentence in the discussion and suggested that a “mild” steatosis was achieved in MS group (lines 307 and 308).

Folch J, Lees M, Sloane Stanley G. A simple method for the isolation and purification of total lipides from animal tissues. J Biol Chem 1957; 226(1):497-509.

Lasa A, Simón E, Churruca I, Fernández-Quintela A, Macarulla MT, Martínez JA, Portillo MP. Effects of trans-10,cis-12 CLA on liver size and fatty acid oxidation under energy restriction conditions in hamsters. Nutrition. 2011 Jan;27(1):116-121. doi: 10.1016/j.nut.2010.01.003. Epub 2010 Aug 8.

Qasem RJ, Li J, Tang HM, Browne V, Mendez-Garcia C, Yablonski E, Pontiggia L, D'Mello AP. Decreased liver triglyceride content in adult rats exposed to protein restriction during gestation and lactation: role of hepatic triglyceride utilization. Clin Exp Pharmacol Physiol. 2015 Apr;42(4):380-8. doi: 10.1111/1440-1681.12359.

Nassir, F.; Rector, R.S.; Hammoud, G.M.; Ibdah, J.A. Pathogenesis and Prevention of Hepatic Steatosis. Gastroenterol Hepatol (N Y) 2015, 11, 167-175. 

2)      Table 1 shows a body weight increase by the snacks fed rats than the MetS group. Although they mentioned it statistically not significant, an increase of approximately 25-30g in a 4 week period would be a serious concern. Hence it needs to be justified.

As previously mentioned, MS rats were sacrificed after 8 weeks of high-fat high-fructose diet feeding and snacks fed rats were sacrificed 4 weeks later. Thus, the body weight increase in snack treated rats was due to a longer treatment period.

MS rats were sacrificed at 8th week of the experiment, in order to settle up the NAFLD starting point. Afterward, the treatment period started for the rest of animal groups WC, OC, SA and SB groups. Although fourteen-week-old rats are mature animals (6 sex-week-old + 8 week of feeding high-fat high-fructose diet,) they continue growing.  Due the different age at time of sacrifice (14 week-old for MS rats and 18 week-old for the rest of rats), MS rats weighed less than the rest. 

With the intention to provide a more detailed information about the experiment period, a flow chart illustrating the experimental design has been included as supplementary figure 1 in the revised version.

3)      Statistical comparison between the groups is confusing. I do not understand what “a” and “b” denotes in all the figures. The authors need to indicate in figure legends.

When statistical analysis was performed in the present study, one-way ANOVA followed by Newman-Keuls post hoc test was used in order to detect differences among groups. “a”, “b” and “c” letters mean differences between groups (P<0.05). “a” letter is given to the group with the highest value and “b” and “c” letters to the ones that show lower values. The meaning of different letters has been explained in each figure legend. However, it is true that higher differences (P<0.01 or P<0.001), have not been specifically indicated in the manuscript. Thus, and following the referee's suggestion, when P value was lower than 0.01 or 0.001 an specific sentence has been added in the figure legends of the revised version (see figure legends of Table 1, Table 2, Figure 2, Figure 3 and Figure 5 in the revised version).

4)      Results presented in Fig 5B-C need to be discussed

As no changes were observed in the protein expression of FATP2 and FATP, transporters involved in fatty acid uptake, authors considered not to discuss them in depth. However, and according to the referee´s comment, a more specific mention to these results has been included in the Discussion of the revised manuscript (line 322). 

5)      There is no explanation or discussion of the results presented in Fig 6A-C

A sentence for the explanation of results presented in Figures 6A, B and C has been written in the revised version (lines 322-323).

Round  2

Reviewer 1 Report

The revised manuscript has been improved significantly. The major limitation of this study is unavailbility of liver tissue samples for any kind of histopathology, steatotic marker (oil redO) or immunohistochemistry imaging which author has noted in revised manuscript. This is not clear why wheat control (WC) data is not provided in Fig 2 and further figures, however most of the figure legend mention about the WC. Author should provide logical explanation of removing this control group and accordingly revise the manuscript.  

Author Response

Reviewer 1

The revised manuscript has been improved significantly. The major limitation of this study is unavailbility of liver tissue samples for any kind of histopathology, steatotic marker (oil redO) or immunohistochemistry imaging which author has noted in revised manuscript. This is not clear why wheat control (WC) data is not provided in Fig 2 and further figures, however most of the figure legend mention about the WC. Author should provide logical explanation of removing this control group and accordingly revise the manuscript.

Wheat control group was fed with a normocaloric diet where the major source of carbohydrates was wheat starch. This group was included in the study in order to simulate a healthy and equilibrate dietary feeding pattern. By contrast, as the snacks were formulated with oat flour, authors considered that a proper control group for the treated groups should have oat flour starch in the diet. Thus, OC group was included in the study.

Liver composition and serum determinations (Table 1, Table 2 and Figure 1) were carried out in all the experimental groups because the comparison with both control groups was interesting. The comparison with OC group clarified whether the effects observed in the snack treated groups were due to the oat or to the functional ingredients present in the snacks (carob and wakame). On the other hand, comparison with WC group clarified whether the snack consumption vs. following an equilibrate diet was better or not. When mechanisms of action were analyzed (Western Blot and PCR), we only wanted to know which were the mechanisms for liver TG reduction after snack treatment. Thus, in these assays only a possible oat effect needed to be discarded and thus, only OC group was included.

Following the referee´s suggestion, an explanation has been included in the discussion section of the revised version (lines 317-319).

Reviewer 3 Report

The revisions performed are relevant and made the manuscript interesting to the wide range of readers in the field metabolic syndrome